# Hydroxylated Coumarin-Based Thiosemicarbazones as Dual Antityrosinase and Antioxidant Agents

**DOI:** 10.3390/ijms24021678

**Published:** 2023-01-14

**Authors:** Sebastiano Masuri, Benedetta Era, Francesca Pintus, Enzo Cadoni, Maria Grazia Cabiddu, Antonella Fais, Tiziana Pivetta

**Affiliations:** 1Department of Chemical and Geological Sciences, University of Cagliari, S.P. 8 km 0.700, 09042 Cagliari, Italy; 2Department of Life and Environmental Sciences, University of Cagliari, S.P. 8 km 0.700, 09042 Cagliari, Italy

**Keywords:** tyrosinase, coumarin, melanogenesis, enzyme inhibition

## Abstract

The design of novel antityrosinase agents appears extremely important in medical and industrial sectors because an irregular production of melanin is related to the insurgence of several skin-related disorders (e.g., melanoma) and the browning process of fruits and vegetables. Because melanogenesis also involves a nonenzymatic oxidative process, developing dual antioxidant and antityrosinase agents is advantageous. In this work, we evaluated the antioxidant and tyrosinase inhibition ability of two new bishydroxylated and two new monohydroxylated derivatives of (1*E*)-2-(1-(2-oxo-2*H*-chromen-3-yl)ethylidene)hydrazine-1-carbothioamide (**T1**) using different experimental and computational approaches. The study was also carried out on another monohydroxylated derivative of **T1** for comparison. Interestingly, these molecules have more potent tyrosinase-inhibitory properties than the reference compound, kojic acid. Moreover, the antioxidant activity appears to be influenced according to the number and substitution pattern of the hydroxyl groups. The safety of the compounds without (**T1**), with one (**T3**), and with two (**T6**) hydroxyl groups, has also been assessed by studying their cytotoxicity on melanocytes. These results indicate that (1*E*)-2-(1-(2-oxo-2H-chromen-3-yl)ethylidene)hydrazine-1-carbothioamide and its hydroxylated derivatives are promising molecules for further drug development studies.

## 1. Introduction

Melanin, the pigment mainly responsible for skin colouration, is a heterogenous biopolymer produced in melanocytes through a well-known process called melanogenesis. Tyrosinase (EC 1.14.18.1) is a key enzyme for melanin production due to its involvement in the rate-limiting step of melanogenesis. This enzyme catalyses the oxidation of phenols and diphenols thanks to the presence of a binuclear copper-based catalytic site where the metal ions are coordinated by six histidine residues. During melanogenesis, tyrosinase is involved in the biosynthesis of dopaquinone, which can take place directly from L-tyrosine through the monophenolase catalytic cycle or passing through the hydroxylation of the same substrate to obtain L-dihydroxyphenylalanine (L-DOPA) by means of the diphenolase cycle [1]. Even if several tyrosinases from microorganisms, plants, and animals have been sequenced, the complete characterisation concerns only a few of them. Tyrosinase structure is characterised by three domains called central, N-terminal, and C-terminal. The central domain is the most conserved domain and comprises the catalytic site.

The development of novel tyrosinase inhibitors is of great importance for both medical and technological applications. Indeed, an irregular expression of tyrosinase determines the accumulation of melanin, which is related to the insurgence of several skin pigmentation disorders, such as melasma, senile spots, freckles, and even malignant melanoma tumours [1,2]. Tyrosinase is also involved in the browning process of fruits and vegetables, and tyrosinase inhibitors could therefore find potential applications as food preservatives as well [3].

Many natural coumarins, such as umbelliferone (7-hydroxy-2-*H*-chromen-2-one), have proved to act as tyrosinase inhibitors [4]. Thanks to both their synthetic accessibility and higher versatility in terms of biological properties, which aretuneable according to the nature and position of the substituents in the coumarinic skeleton [5], many scientists have evaluated the antityrosinase activity of several coumarinic derivatives [1,6]. The insertion of hydroxyl groups in the structure of 3-aryl and heteroarylcoumarins led to novel antioxidant compounds with potent tyrosinase inhibitory properties [7,8,9]. Considering that melanogenesis also involves a nonenzymatic oxidative process, the development of novel compounds with dual antioxidant and tyrosinase inhibitory properties appears quite useful [10,11].

Many molecules bearing a thiosemicarbazone moiety proved to possess promising antityrosinase properties [12]. Based on that, we aimed to investigate the dual antioxidant and antityrosinase activity of a series of hydroxylated derivatives of the inhibitor (1*E*)-2-(1-(2-oxo-2*H*-chromen-3-yl)ethylidene)hydrazine-1-carbothioamide (**T1–T6**, Figure 1) [13] using different experimental and computational approaches. By combining experimental (protonation constants) and theoretical results (molecular descriptors), the drug-likeness of the studied compounds and their speciation in different biological compartments have been evaluated. The safety of the studied molecules has also been assessed by studying their cytotoxicity on melanocytes.

## 2. Results and Discussion

### 2.1. Synthesis and Chemical Characterization

The target compounds **T1–6** were successfully obtained according to the two-step synthetic strategy outlined in Figure 1. In brief, salicylaldehyde and its hydroxylated derivatives were subjected to Knoevenagel condensation in the presence of ethyl acetoacetate (as active methylene compound) and piperidine (as base) to afford the 3-acetylcoumarins **1–6**. The former intermediates were easily converted in the corresponding thiosemicarbazones by reacting them with thiosemicarbazide in the presence of acetic acid as a catalyst. The yields of the **T1–6** target compounds ranged from moderate to good. The structure and purity of both the intermediates and final compounds were confirmed with ^1^H NMR, elemental analysis, and MS data. The NMR spectra for **5** and **T4–T6** are reported in the Supporting Appendix A.

### 2.2. Antioxidant Assay

Radical scavenging activities were evaluated by using 1,1-diphenyl-2-picrylhydrazyl free radical (DPPH) and 2,2′-azinobis-(3-ethylbenzothiazoline-6-sulfonic acid (ABTS) radical scavenging assays [14,15,16].

The antioxidant activity of **T1** and its hydroxylated derivatives **T2–6** was evaluated via DPPH assay, which is based on the ability of the stable DPP· radical species to be converted in its diamagnetic form (DPPH) by accepting an H· from the tested compound. In ethanol, the DPP· radical shows an intense violet colouration (with a λ_max_ at 517 nm), whereas the DPPH species is yellow-coloured in the same solvent. From the evaluation of the relative decrease in absorbance at different concentrations of each tested compound, the antioxidant activities were evaluated by reporting their EC_50_ values, as shown in Table 1.

In comparison with ascorbic acid (used as reference), compounds **T1–3** were devoid of any radical scavenging capability, whereas in **T4**, where an –OH was present in the eighth position of the coumarinic scaffold, modest antioxidant activity was observed. On the contrary, compounds **T5** and **T6**, both having a catechol-like motif, showed the most potent antioxidant properties. The different EC_50_ values achieved for these two compounds (7.1 and 17.9 µM) suggest that the relative position of the catecholic motif in the **T1** structure plays an important role in defining the antioxidant activity of this panel of compounds (**T5** and **T6** were, in fact, 6,7 and 7,8 dihydroxylated, respectively).

The ABTS test confirmed the antioxidant capacity of **T5** and **T6** compounds with EC_50_ values lower than that of the reference compound (Trolox). The antioxidant activities, expressed as EC_50_ values, are shown in Table 1 for all of the compounds.

### 2.3. Tyrosinase Inhibition Assay

The antityrosinase activity of the studied compounds was preliminarily evaluated using mushroom tyrosinase as a protein model. Results from IC_50_ values (Table 2) show that all of the compounds (except **T2**) possessed higher tyrosinase inhibitory activity compared to the known tyrosinase inhibitor, kojic acid [9,17].

Kinetics parameters for compounds without hydroxyl (**T1**) and with two hydroxyl (**T6**) groups were determined. The Lineweaver–Burk plot for compound **T1** shows that V_max_ was the same regardless of the inhibitor concentration, whereas Km increased with the inhibitor concentrations (Figure 2A). Therefore, compound **T1** exhibited competitive inhibition with a Ki of 0.85 μM.

When hydroxyl groups were present in the coumarin ring (positions 7 and 8), the inhibition potency was maintained, and it exhibited mixed-type inhibition (Figure 2B).

This behaviour shows that compound **T6** can bind not only with the free enzyme but also with the enzyme–substrate complex, and that the related equilibrium constants are different. Accordingly, the inhibition constants for the inhibitor binding with free enzyme (Ki) and enzyme–substrate complex (Kis) resulted in 5.88 and 10.65 μM, respectively. The secondary plots to determine the inhibition constants are shown in Appendix A.

### 2.4. Drug-Likeness and Biospeciation Studies

Preliminary evaluation of the drug-likeness of **T1–6** has been carried out using different molecular descriptors, as shown in Table 3. All of the compounds adhered to Lipinski’s rule of five, which states that poor absorption or permeation is commonly experienced in the presence of more than five hydrogen bond donors, 10 hydrogen bond acceptors when the LogP is higher than 5, and the molecular weight (MW) is greater than 500 Dalton [18]. The drugs’ ability to permeate biological membranes could also be evaluated based on their Topological Polar Surfae Area (TPSA) values, which are defined as the sum of the surface occupied by polar functional groups [19]. The **T1–6** compounds had TPSA values ranging from 80.62 (**T1**) to 121.08 Å^2^ (**T5** and **T6**), which is indicative of good oral bioavailability because they were below the upper limit of 140 Å^2^. All of the studied compounds possessed TPSA values higher than 60 Å^2^, which is usually associated with compounds having a modest blood–brain barrier (BBB)-crossing ability [20].

When a molecule has ionisable groups (e.g., -RR’NH, -OH, -CO_2_H, etc.), both its physicochemical properties and bioavailability could be modified according to the pH of the solution. Moreover, the observed pharmacological activity may depend on the real present species, such as L or H_n_L^n+^. This aspect should be considered in the design and synthesis of novel bioactive molecules because they could exist in differently protonated and charged forms according to the pH of the biological fluid or loading compartment (stomach, intestine, blood, etc.) where the compound is located.

The biospeciation of the studied molecules, i.e., to define which species is present in a specific bioenvironment, could be achieved by combining the pH distribution curves obtained via solution equilibria studies with the acidity and basicity values of different biological fluids or compartments.

The pH distribution curves can be obtained with the knowledge of the protonation constants of the molecules. Solution equilibria studies allow us to determine the protonation constants of a compound bearing ionisable functional groups. The protonation constants of **T1–6** have been determined by simultaneous potentiometric and spectrophotometric titrations (pH range: 3–11). Selected spectra, recorded during the titrations, are shown in Figure 3. More details are given in Appendix A. Potentiometric and spectrophotometric results allowed us to calculate the cumulative protonation constants (Table 4) and to identify the pK values. Taking **T1** and **T5** as examples, the spectral variations observed during their titrations are reported in depth in Appendix A. Eigenvalue analysis on the spectrophotometrical data gave four linearly independent absorbing species for **T1** ($H_{3}L^{2+}$, $H_{2}L^{+}$, HL, $L^{-}$), five for **T2–4** ($H_{4}L^{2+}$, $H_{3}L^{+}$, $H_{2}L,$ ${HL}^{-}$, $L^{2-}$), and six for compounds **T5** and **T6** ($H_{5}L^{2+}$, $H_{4}L^{+}$, $H_{3}L$, ${H_{2}L}^{-}$, ${HL}^{2-}$, $L^{3-}$). The simultaneous analysis of both potentiometric and spectrophotometrical data allowed us to obtain the cumulative protonation constants and their related pK values (Table 4). Absorptivity values, reported at maximum wavelengths, for all of the absorbing species are reported in Appendix A, whereas their calculated pure spectra are shown in Appendix A.

Considering two different mediums, such as blood plasma and the gastrointestinal (GI) tract, we can observe that regarding the blood plasma at (pH 7.4), **T1** is present in its neutral form (as HL), whereas the other compounds mainly exist in their monoanionic forms (HL^−^ for **T2–4** and H_2_L^−^ for **T5** and **T6**). The situation is more complicated in the GI tract because its local pH values vary among individuals, according to factors like food ingestion, age, presence of systemic diseases, and concomitant drug administration [21]. In the intestinal compartments, the pH fluctuates between 4.9 and 7.4 at the fasted state and 5.2–7.5 at the fed state [22]. In any case, the combined use of these average values and the experimental data at our disposal led to a preliminary evaluation of the region of the GI tract where these molecules are absorbed in their neutral forms via a passive diffusion mechanism. In the GI tract, compound **T1** would be predominantly absorbed in the ileum region, where the pH fluctuates between six and eight, whereas the other molecules would be more absorbed in the duodenum (4–6 pH range). The same compounds could be absorbed in the stomach during the fed state.

### 2.5. Cell Viability

The cytotoxicity of compound **T6** (Figure 4), which showed the strongest tyrosinase inhibition and good antioxidant activity, was measured to determine the safety of this molecule.

At 4 μM, the concentration in which the compound inhibited tyrosinase activity, no cytotoxic effect in B16F10 cells was observed. At the highest concentration tested, 100 μM, 25-fold higher than the IC_50_ values, the cell viability of the compound was still greater than 90%. The same protocol was applied for compounds without hydroxyl (**T1**) and with one hydroxyl (**T3**) groups, obtaining similar results (Appendix A).

### 2.6. Molecular Docking


**Input ligands structures.**


Molecular docking simulations between mushroom tyrosinase (PDB: 2Y9X, A-chain) and the **T1–6** molecules were performed considering the pH levels where the enzymatic inhibitory studies have been carried out (pH 6.8). As observed, at pH 6.8, compounds **T1–4** existed in their neutral forms as HL for **T1** and $H_{2}L$ for **T2–4**, whereas **T5** and **T6** were mainly present in their monoanionic form, ${H_{2}L}^{-}$. The structures of these species were simulated in water using the CPCM solvation model from DFT calculations. The results are shown in Appendix A. The structural metrics (bond lengths, angles, and dihedrals) of the DFT-optimized structure of **T1** (as HL) were consistent with the X-ray structure of the same compound [23] as evident from the superimposition of the structures, where the root-mean-square deviation (RMSD) was 0.12 Å (Appendix A). In Appendix A, selected structural parameters for both calculated and crystallographic structures are reported. The structures of the monoanionic species ${H_{2}L}^{-}$ for compounds **T5** and **T6** were assessed by determining the most plausible site of deprotonation through the calculation of the proton affinity (PA) values in the gas phase and water. The results, summarised in Appendix A, indicate that compound **T5** lost the acidic hydrogen from the hydroxyl group in the sixth position, whereas **T6** released the proton from the -OH in the eighth one. This feature is related to the presence of a catechol-like motif in both molecules, which brings the formation of an intramolecular hydrogen bond between the phenate anion and the hydrogen of the neighbouring -OH groups (ortho effect) (Appendix A).


**Molecular docking protocol validation.**


The docking protocol adopted has been validated by redocking the cognate OTR ligand in the receptor’s binding site. The molecular interactions between the docked pose of OTR and the surrounding residues of the receptor have been compared with those established by the crystallographic structure of the same ligand. The results are shown in Appendix A. The docked pose of OTR kept the π-π interactions with His263 and Val283, as observed in the crystallographic structure. On the contrary, the interaction between His259 and the X-ray structure of the ligand has been replaced by two new hydrogen bonds formed by the residues of His85 and Val283 with the docked OTR molecule. This difference could be attributed to two factors: (*i*) the resolution of the selected X-ray structure (2.78 Å); (*ii*) the fact that the scoring functions do not consider either entropy factors or solvation effects (simulations are performed in the gas phase) that might take place upon interaction between the receptor and the ligand. Assessment of the docking protocol performance is commonly performed by calculating the RMSD of the atomic positions of the docked pose with the X-ray structure of the same ligand. The results, reported in Appendix A, show an adequate overlapping (RMSD: 2.32 Å) between the predicted and experimental structures of the cognate OTR ligand.


**Molecular docking simulation results.**


The validated protocol has been applied for docking the studied thiosemicarbazones in the catalytic site (MTa) of mushroom tyrosinase. Taking compounds **T1** and **T6** as examples (Figure 5 and Figure 6), it is possible to observe how the presence of -OH substituents significantly alters their binding orientations in the receptor’s active site. In the case of **T1** (docking score: 38.8535), the ligand was oriented with the thiosemicarbazone moiety towards the catalytic site, whereas the coumarinic ring was directed towards the enzyme surface. This molecule can coordinate both copper ions of the catalytic site using its thiocarbonyl group, thus suggesting the formation of a ternary complex with these metal cofactors as a potential mechanism of inhibition. The predicted pose was further stabilised by the presence of different intermolecular interactions with the surrounding receptor residues (e.g., π-π T-shaped with Phe264, π-alkyl with Val248, His259, π-sigma with His263, and π-Sulphur with His61, 259, and 296). In the case of compound **T6** (docking score: 52.5943), the insertion of -OH groups in the coumarinic backbone resulted in a 180° twist of the predicted pose, with the hydroxyl groups oriented towards the enzyme’s catalytic site and the thiosemicarbazone pointing towards the enzyme’s surface. As observed for the other hydroxylated thiosemicarbazone derivatives, this conformation was stabilized by the presence of the π-alkyl and π-sigma interaction between the Val283 residue and the benzopyran-2-one ring of the ligands. In the specific case of **T6**, the docked pose was further stabilized by additional intermolecular interactions with the surrounding residues (e.g., H-bonds with His94, π-alkyl with Phe264, π-π stacked with His263, and π-π T-shaped with His85). This molecule can potentially coordinate the Cu401 ion through its oxygen atoms.

According to Lineweaver–Burk plots, compound **T6** acts as a mixed-type inhibitor towards mushroom tyrosinase. Hassani et al. have shown by means of theoretical methods that the mixed inhibitors phthalic and cinnamic acids could exert their mode of action by interacting with two different allosteric sites (MTb for phthalic acid and MTc for cinnamic acid) in the complex between mushroom tyrosinase and tropolone [24]. Based on that, we evaluated the potential interaction of **T6** at both allosteric sites by means of molecular docking. In Appendix A, the highest-ranking docking poses of **T6** at both allosteric sites are summarized. A closer inspection of the intermolecular interactions reveals, for the docked pose of **T6** in the MTb site, π-alkyl interactions with Pro366, Pro370, and Trp386 as well as π-π stacked interactions with His390 (Appendix A). Regarding the docked pose of **T6** in the MTc site, the molecule was stabilized by π- π interactions with Trp358 and C-hydrogen bond donor with Thr308 (Appendix A). Considering the closeness of the docking score values obtained (43.4042 and 38.2733 for **T6** when docked in MTb and MTc sites, respectively), it could be assumed that this mixed-type inhibitor might form ternary complexes with the enzyme and the substrate by binding at one of these allosteric sites.

## 3. Materials and Methods

### 3.1. Synthesis and Characterization Techniques

Ethyl acetoacetate, DPPH, ABTS, deuterated dimethyl sulfoxide, and salicylaldehyde and its hydroxylated derivatives were purchased from Alfa Aesar (Kandel, Germany). Absolute ethanol, piperidine, deuterated chloroform, thiosemicarbazide, and acetic acid were purchased from Merck (Milan, Italy). Reagents were used as received without further purification.

NMR spectra were acquired with a Bruker Advance III HD 600 spectrometer (Rheinstetten, Germany) at room temperature with tetramethylsilane (TMS) as the internal standard in DMSO-d6 or CDCl_3_. ESI mass spectra were recorded with a triple quadrupole QqQ Varian 310-MS mass spectrometer (Palo Alto, CA, USA) using previously optimized parameters [5]. High-resolution ESI mass spectra were acquired on a Thermofisher ORBITRAP-ELITE (Waltham, MA, USA). Melting points were measured on a Kofler Hot Stage (Rochford, UK) and are uncorrected.

### 3.2. General Procedure for the Synthesis of Substituted 3-Acetyl-2H-chromen-2-one Derivatives

Salycilaldehyde derivatives (7.24 mmol, 1.0 eq) and ethylacetoacetate (8.64 mmol, 1.2 eq) were dissolved in 2.6 mL of ethanol, and piperidine was then added (0.15 mmol, 0.02 eq). The reaction mixture was left under stirring at reflux for 5 h. For compounds **1–4** and **6,** the reaction mixture was cooled to room temperature, affording a precipitate that was filtered and recrystallized from methanol. For compound **5**, the reaction mixture was poured into cold water and treated with HCl 10% until an acidic pH level was reached. A dark brown solid was obtained after a few minutes of stirring. The solid was filtered, dried, repeatedly washed with Et_2_O, and used as such for the next step.

**3-acetyl-2*H*-chromen-2-one (1).** Light yellow solid. Yield 81%. Experimental results are in accordance with those reported in the literature [13]. Melting point: 178–180 °C. ^1^H-NMR (600 MHz, CDCl_3_, δ, ppm): 8.51 (d, J = 0.7 Hz, 1H), 7.69–7.63 (m, 2H), 7.40–7.32 (m, 2H), 2.73 (s, 3H). ESI-MS (calculated, found, *m*/*z*): 211.0, 211.0 [M+Na]^+^.

**6-hydroxy-3-acetyl-2*H*-chromen-2-one (2).** Yellow solid. Yield 63%. Experimental results are in accordance with those reported in the literature [25,26]. Melting point: 244–245 °C. ^1^H-NMR (600 MHz, DMSO d6, δ, ppm): 9.89 (s, 1H), 8.57–8.54 (m, 1H), 7.31 (d, J = 8.9 Hz, 1H), 7.24 (d, J = 2.9 Hz, 1H), 7.17 (dd, J = 8.9, 2.9 Hz, 1H), 2.58 (s, 3H). ESI-MS (calculated, found, *m*/*z*): 203.0, 203.0 [M-H]^−^.

**7-hydroxy-3-acetyl-2*H*-chromen-2-one (3).** Light brown solid. Yield 34%. Experimental results are in accordance with those reported in the literature [26,27]. Melting point: 243–244 °C. ^1^H-NMR (600 MHz, DMSO d6, δ, ppm): 11.10 (br. s, 1H), 8.59 (d, J = 0.7 Hz, 1H), 7.78 (d, J = 8.6 Hz, 1H), 6.85 (dd, J = 8.6, 2.3 Hz, 1H), 6.75 (dd, J = 2.3, 0.7 Hz, 1H), 2.55 (s, 3H). ESI-MS (calculated, found, *m*/*z*): 203.0, 203.1 [M-H]^−^.

**8-hydroxy-3-acetyl-2*H*-chromen-2-one (4).** Brown solid. Yield 57%. Experimental results are in accordance with those reported in the literature [26,28]. Melting point: 253–255 °C.^1^H-NMR (600 MHz, DMSO d6, δ, ppm): 10.35 (s, 1H), 8.59 (s, 1H), 7.36 (dd, J = 5.8, 3.4 Hz, 1H), 7.24–7.18 (m, 2H), 2.59 (s, 3H). ESI-MS (calculated, found, *m*/*z*): 205.0, 204.9 [M+H]^+^.

**6,7-dihydroxy-3-acetyl-2*H*-chromen-2-one (5).** Dark brown solid. Yield 61%. Melting point: 192–194 °C. ^1^H-NMR (600 MHz, DMSO d6, δ, ppm, Appendix A): 10.87 (br. s, 1H), 9.67 (br. s, 1H), 8.54 (s, 1H), 7.21 (s, 1H), 6.79 (s, 1H), 2.54 (s, 3H). ESI-MS (calculated, found, *m*/*z*): 219.0, 219.0 [M-H]^−^_._

**7,8-dihydroxy-3-acetyl-2*H*-chromen-2-one (6).** Brown solid. Yield 25%. Experimental results are in accordance with those reported in the literature [26,29]. Melting point: 262–264 °C. ^1^H-NMR (600 MHz, DMSO d6, δ, ppm): 8.53 (s, 1H), 7.29 (d, J = 8.5 Hz, 1H), 6.85 (d, J = 8.5 Hz, 1H), 2.56 (s, 3H). ESI-MS (calculated, found, *m*/*z*): 219.0, 219.0 [M-H]^−^.

### 3.3. General Procedure for the Synthesis of Substituted (1E)-2-(1-(2-oxo-2H-chromen-3-yl)ethylidene)hydrazine-1-carbothioamide Compounds [13]

The proper 3-acetylcoumarin derivative (2.5 mmol, 1.0 eq) was dissolved in 10 mL of ethanol, and then thiosemicarbazide (2.5 mmol, 1.0 eq) and acetic acid (0.12 mL) were added. The reaction mixture was left under stirring at reflux for 6 h, and then it was cooled to room temperature, affording a precipitate that was filtered and recrystallized from ethanol.

**(1*E*)-2-(1-(2-oxo-2*H*-chromen-3-yl)ethylidene)hydrazine-1-carbothioamide (T1).** Yellow solid. Yield: 98%. Experimental data are consistent with those reported in the literature [13]. Melting point: 196–198 °C. ^1^H-NMR (600 MHz, DMSO d6, δ, ppm): 10.44 (s, 1H), 8.48 (s, 1H), 8.40 (s, 1H), 7.95 (s, 1H), 7.77 (dd, J = 7.7, 1.6 Hz, 1H), 7.65 (ddd, J = 8.7, 7.3, 1.6 Hz, 1H), 7.44 (d, J = 8.3 Hz, 1H), 7.40 (td, J = 7.5, 1.1 Hz, 1H), 2.26 (s, 3H). HR ESI-MS (calculated, found, *m*/*z*): 262.0650, 262.0646 [M+H]^+^. Elemental analysis exp(calc.) % C 55.24 (55.16), H 4.32 (4.24), N 16.11 (16.08), S 12.45 (12.27).

**(1*E*)-2-(1-(6-hydroxy-2-oxo-2*H*-chromen-3-yl)ethylidene)hydrazine-1-carbothioamide (T2).** Yellow solid. Yield: 82%. Melting point: 222–224 °C (with decomposition). ^1^H-NMR (600 MHz, DMSO d6, δ, ppm, Appendix A): 10.73 (s, 1H), 10.34 (s, 1H), 8.39 (s, 1H), 8.34 (s, 1H), 7.90 (s, 1H), 7.59 (d, J = 8.6 Hz, 1H), 6.84 (dd, J = 8.5, 2.3 Hz, 1H), 6.74 (d, J = 2.2 Hz, 1H), 2.24 (s, 3H). ^13^C-NMR (151 MHz, DMSO d6, δ, ppm, Appendix A): 179.55, 162.37, 159.99, 155.98, 147.05, 143.10, 131.06, 121.50, 114.15, 111.90, 102.23, 16.50. HR ESI-MS (calculated, found, *m*/*z*): 278.0599, 278.0591 [M+H]^+^. Elemental analysis exp(calc.) % C 51.78 (51.98), H 3.98 (4.00), N 15.18 (15.15), S 11.45 (11.56).

**(1*E*)-2-(1-(7-hydroxy-2-oxo-2*H*-chromen-3-yl)ethylidene)hydrazine-1-carbothioamide (T3).** Experimental data are consistent with those reported in the literature [30]. Yellow-brown earth was solid. Yield: 80%. Melting point: 222–224 °C (with decomposition). ^1^H-NMR (600 MHz, DMSO d6, δ, ppm): 10.40 (s, 1H), 9.80 (s, 1H), 8.37 (br. s, 1H), 8.36 (s, 1H), 7.95 (s, 1H), 7.27 (d, J = 8.0 Hz, 1H), 7.09–7.04 (m, 2H), 2.24 (s, 3H). ^13^C-NMR (151 MHz, DMSO d6, δ, ppm): 179.69, 159.84, 154.36, 147.27, 146.73, 142.41, 126.28, 120.98, 119.92, 117.33, 113.33, 16.56. HR ESI-MS (calculated, found, *m*/*z*): 278.0599, 278.0594 [M+H]^+^. Elemental analysis exp(calc.) % C 52.02 (51.98), H 4.06 (4.00), N 15.16 (15.15), S 11.45 (11.56).

**(1*E*)-2-(1-(8-hydroxy-2-oxo-2*H*-chromen-3-yl)ethylidene)hydrazine-1-carbothioamide (T4).** Light yellow solid. Yield: 93%. Melting point: 208–210 °C (with decomposition). ^1^H-NMR (600 MHz, DMSO d6, δ, ppm, Appendix A): 10.41 (s, 1H), 10.26 (s, 1H), 8.41 (s, 1H), 8.38 (s, 1H), 7.95 (d, J = 3.0 Hz, 1H), 7.19 (d, J = 4.4 Hz, 2H), 7.12 (t, J = 4.7 Hz, 1H), 2.26 (s, 3H). ^13^C-NMR (151 MHz, DMSO d6, δ, ppm, Appendix A): 179.71, 159.53, 146.64, 144.83, 142.90, 142.53, 126.12, 125.17, 120.35, 119.55, 119.12, 16.54. HR ESI-MS (calculated, found, *m*/*z*): 278.0599, 278.0595 [M+H]^+^. Elemental analysis exp(calc.) % C 51.92 (51.98), H 4.06 (4.00), N 15.12 (15.15), S 11.55 (11.56).

**(1*E*)-2-(1-(6,7-dihydroxy-2-oxo-2*H*-chromen-3-yl)ethylidene)hydrazine-1-carbothioamide (T5).** Dark brown solid. Yield: 44%. Melting point: 228–230 °C (with decomposition). ^1^H-NMR (600 MHz, DMSO d6, δ, ppm, Appendix A): 10.38 (s, 1H), 10.30 (s, 1H), 9.52 (s, 1H), 8.32 (s, 2H), 7.90 (s, 1H), 7.04 (s, 1H), 6.77 (s, 1H), 2.23 (s, 3H). ^13^C-NMR (151 MHz, DMSO d6, δ, ppm, Appendix A): 179.50, 160.29, 151.76, 149.28, 147.42, 143.68, 143.05, 121.51, 113.17, 111.42, 102.69, 15.92. HR ESI-MS (calculated, found, *m*/*z*): 294.0548, 294.0545 [M+H]^+^. Elemental analysis exp(calc.) % C 49.24 (49.14), H 3.82 (3.78), N 14.30 (14.33), S 10.89 (10.93).

**(1*E*)-2-(1-(7,8-dihydroxy-2-oxo-2*H*-chromen-3-yl)ethylidene)hydrazine-1-carbothioamide (T6).** Dark yellow solid. Yield: 45%. Melting point: 228–230 °C (with decomposition). ^1^H-NMR (600 MHz, DMSO d6, δ, ppm, Appendix A): 10.32 (s, 1H), 9.42 (br. s, 1H), 8.34 (s, 1H), 8.32 (s, 1H), 7.90 (s, 1H), 7.10 (d, J = 8.5 Hz, 1H), 6.85 (d, J = 8.5 Hz, 1H), 2.25 (s, 3H). ^13^C-NMR (151 MHz, DMSO d6, δ, ppm, Appendix A): 179.52, 159.98, 150.80, 147.28, 144.12, 143.69, 132.30, 121.27, 120.40, 113.50, 112.69, 16.58. HR ESI-MS (calculated, found, *m*/*z*): 294.0548, 294.0546 [M+H]^+^. Elemental analysis exp(calc.) % C 49.09 (49.14), H 3.77 (3.78), N 14.28 (14.33), S 10.90 (10.93).

### 3.4. Potentiometric and Spectrophotometric Titrations

Potentiometric titrations were performed in a thermostated vessel at 25 °C in 0.1 M ionic strength (NaCl) using a Mettler-Toledo Seven Compact pH/Ion-meter and equipped with a Mettler-Toledo InLab Micro Pro combined glass electrode with an integrated temperature probe. The glass electrode was calibrated daily and checked using the GLEE software package [31]. UV–visible (UV–Vis) measurements were carried out with an Agilent Cary 60 spectrophotometer using a 0.1 cm quartz cuvette. Protonation constants of the studied molecules were determined via spectrophotometric and potentiometric titrations. Solutions of ligands were prepared daily by dissolving a suitable amount of the compound in DMSO (concentration ≈ 12 mM) prior to the required dilution in 0.1 M NaCl (final DMSO content 1% V/V, concentration ≈ 1.2∙10^−4^ M). Four HCl equivalents for **T1**, five for **T2–4**, and six for **T5–6** were added before titrating with an NaOH standard solution. Potentiometric and spectrophotometric data were simultaneously analysed using the Hyperquad 2006 software package [32]. Speciation diagrams were obtained using Hyss 2009 program [33] with the spectrophotometric data. The number of linearly independent absorbing species was found by applying the eigenvalue analysis to the absorbance data matrix.

### 3.5. DFT Calculations

DFT calculations were performed on an Intel i7-based system using the release 4.2.0 of the ORCA software package [34]. Input files for DFT calculations were prepared using Avogadro 1.2.0 [35]. Geometry optimisations were performed using the PBE0 functional [36] and def-2 TZVP basis set [37]. Geometry optimisations were performed starting from structural data when available. The nature of the minima obtained after each optimisation was verified by assessing the absence of negative calculated IR frequencies derived from the calculations of the Hessian matrix. DFT studies were performed both in the gas phase and in the presence of a solvent (water) using the conductor-like polarisable continuum model (CPCM) [38].

For compounds **T5** and **T6** ($H_{3}L$), the structures of their monoanionic form have been evaluated by calculating the affinity towards their acidic hydrogens, expressed by the “proton affinity” (PA) thermochemical descriptor [16]:$$\mathrm{PA}=E\left(H_{2}L^{-}\right)+E\left(H^{+}\right)-E\left(H_{3}L\right)$$
where $E\left(H_{2}L^{-}\right)$ is the formation enthalpy of the deprotonated species and $E\left(H^{+}\right)$ and $E\left(H_{3}L\right)$ are the formation enthalpies of the proton and the neutral molecules, respectively. Proton gas-phase enthalpy was taken from the literature as 1.481 kcal mol^−1^ [39], whereas proton water and ethanol enthalpy were calculated by assuming the solvation of a proton using the CPCM model as previously explained [16,40]

### 3.6. Drug-Likeness Descriptors

Molecular descriptors, such as TPSA (topological polar surface area), miLogP (calculated logarithm of the octanol: water partition coefficient), number of hydrogen bond donors and acceptors, number of rotatable bonds, molecular weight, molecular volume, and number of violations of Lipinski’s rule, were calculated using the Molinspiration property engine (v2021.10) (Molinspiration Cheminformatics Free Web Services, https://www.molinspiration.com (accessed on 9 January 2023), Slovensky Grob, Slovakia, n.d.)

### 3.7. Antioxidant Assays

Radical scavenging activities were measured with DPPH and ABTS radical scavenging assays [14,15,16]. For both methods, results are reported as the concentration of antioxidant required to quench 50% of the original absorbance (half maximal effective concentration, EC_50_). In brief, for the DPPH antioxidant assay procedure, each test compound was dissolved in DMSO at 1.0 mM concentration and then diluted with absolute ethanol in the 500–2.0 µM concentration range. A solution of the DPPH radical (0.1 mM, absolute ethanol) was freshly prepared, stored in the dark, and used within a few hours. Each test solution (1500 µL) was vigorously mixed to an equal volume of DPPH inside a cuvette, and the molecular absorbance at 517 nm was recorded for 30 min. Each test was evaluated in three independent experiments. Ascorbic acid was used as the reference compound. For the ABTS method, based on the capacity of an antioxidant to scavenge the free ABTS^+^ generated in the aqueous phase by reacting with a strong oxidizing, the free radical ABTS was produced by reacting 7 mM ABTS with 2.45 mM potassium persulfate in aqueous solution and kept in darkness at room temperature for 24 h before use. Samples of each compound (10 µL) were added to 990 µL of ABTS, and the reduction of the blue-green radical ABTS by hydrogen-donating antioxidants was evaluated by measuring the absorbance at 734 nm after 1 min incubation. The 6-hydroxy-2,5,7,8-tetramethylchromane-2-carboxylic acid (Trolox) was used as the antioxidant standard.

### 3.8. Tyrosinase Inhibition Assay

The inhibition of mushroom tyrosinase was evaluated as previously described [41] with slight modifications. A mixture containing 50 mM phosphoric acid buffer solution (pH 6.8), a solution of mushroom tyrosinase (72 U/mL final concentration; Sigma Chemical Co., Milan, Italy), and DMSO with or without the sample were incubated at 37 °C for 10 min. Then, L-DOPA solution (0.5 mM final) was added, and the reaction was monitored following the molecular absorbance at 492 nm that resulted from the formation of the dopachrome product. Measurements were performed using a FLUOstar OPTIMA (BMG Labtech, Offenburg, Germany). The IC_50_ value, i.e., concentration giving 50% inhibition of tyrosinase activity, was determined by analysing the dose-response curves. Kojic acid was used as a reference tyrosinase inhibitor. The inhibition mode was analysed using the Lineweaver–Burk plots.

### 3.9. Cell Viability Assay

The cellular cytotoxicity of compound **T6** was investigated using a 3-(4,5-dimethylthiazol-2-yl)-2,5-diphenil-tetrazolium bromide (MTT) assay. Murine melanoma B16F10 cells were seeded in 96-well plates at a density of 5 × 10^3^ cells/well and cultured in Dulbecco’s Modified Eagle Medium (DMEM) with 10% fetal bovine serum at 37 °C in a humidified atmosphere with 5% CO_2_. The B16F10 cell line was exposed for 48 h to compound **T6** at concentrations ranging from 4 to 100 μM. After incubation, culture supernatants were removed and exchanged with a medium containing 0.5 mg/mL MTT. Then, after 3 h of incubation at 37 °C, the cells were lysed with 100 μL of DMSO. The absorbance was determined at 560 nm using a microplate reader (Multiskan FC-Thermo Scientific Inc. Waltham, MA, USA).

### 3.10. Molecular Docking

Molecular docking simulations were performed using the CCDC GOLD software (v2022.1.0, Cambridge, UK) [42]. DFT-optimized structures of the ligands were exported as “*.mol2” files for molecular docking simulations. The crystal structure of the complex between mushroom tyrosinase (from *Agaricus bisporus*) and the inhibitor tropolone (OTR) was retrieved from the Protein Data Bank (PDB code: 2Y9X) [43]. The A-chain was used as a receptor for docking studies. The receptor was prepared using the GOLD setup wizard; specifically, the native OTR ligand was extracted from the receptor, the water molecules were removed, and the polar hydrogens were added throughout the enzyme structure. A binding cavity centred at the OTR coordinates that included all residues within 15 Å was chosen for docking simulations in the catalytic site of mushroom tyrosinase (MTa). Each ligand was submitted to 100 genetic algorithm runs using the Chemscore function, followed by rescoring using the ChemPLP function. The “Fitness & Search” options were kept as default. The search efficiency was set to maximum (200%) to comprehensively evaluate the docking conformational space. Validation of the docking protocol has been performed by redocking the cognate OTR ligand into its binding site and calculating theRMSD of the atomic position between the docked and crystallographic poses. For mixed-type inhibitors, molecular docking simulations were also extended to the allosteric sites MTb and MTc, keeping the cognate ligand OTR in the catalytic MTa site as previously done by Hassani et al. [24]. Regarding the MTb site, docking simulations were performed at coordinates x = 11.64, y = −28.74, and z = −31.95. In the case of the allosteric MTc site, a cavity centred at coordinates x = 3.20, y = −10.74, and z = −37.73 was chosen. Molecular interactions and docked poses were evaluated using CCDC Hermes (v2022.1.0, Cambridge, UK) [43] and BIOVIA Discovery Studio Viewer 2019 (Dassault Systèmes BIOVIA, Discovery Studio Viewer, V19, San Diego: Dassault Systèmes, 2019, n.d.)

### 3.11. Statistical Analyses

The evaluation of statistically significant differences was performed by calculating a one-way ANOVA followed by execution of the Bonferroni Multiple Comparisons Test, both using the Graph Pad INSTAT software v8.0 (GraphPad Software, San Diego, CA, USA).

## 4. Conclusions

The results reported in this study show how the insertion of hydroxyl groups in the structure of the coumarin-based thiosemicarbazone **T1** allowed us to obtain a panel of novel compounds with potent antityrosinase activity and tuneable antioxidant properties.

The most promising compound of the series (**T6**) is a mixed-type inhibitor towards tyrosinase, as seen on Lineweaver–Burk plots, with antityrosinase activity higher than that of kojic acid. The safety of **T6** has been proved from in vitro studies on B16F10 cells, where the studied molecule is not cytotoxic even at a higher concentration (25-fold) than its IC_50_ (4.1 μM). Based on molecular docking, this mixed-type inhibitor might target both catalytic (MTa) and allosteric (MTb and MTc) binding sites. Notably, these simulations were performed considering the protonated main species of the ligands really present at the pH experimental conditions chosen for tyrosinase inhibition assay. Thanks to the presence of a catechol-like motif, **T6** showed more potent antioxidant properties than natural ascorbic acid and trolox antioxidants.

Thanks to the multidisciplinary approach proposed here, these encouraging results could be useful in designing novel coumarin-based molecules of biological relevance.

## Data Availability

Data are present within the article.

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
