# Peer review of "Hydroxylated Coumarin-Based Thiosemicarbazones as Dual Antityrosinase and Antioxidant Agents"

_ijms, 2023, doi:10.3390/ijms24021678_

Round 1

Reviewer 1 Report

General Comment:

Results and Discussion could be rearranged, since some of the experimental and computational approaches are applied only to selected compounds, according to their antioxidant and anti-tyrosinase activity.

For example, results for "The safety of the studied molecules has also been assessed by studying their cytotoxicity on melanocytes" stated in the Abstract, are only reported for T1, T3 and T6.

Also, molecular docking simulation results for compounds T2-T5 are not presented.

Therefore, Antioxidant assay and Tyrosinase inhibition assay should precede Cell viability and Drug-likeness and bio-speciation studies.

Moreover, Drug-likeness and bio-speciation studies section could also focus on the most promising compounds.

Specific Questions:

"All the results show that hydroxylated derivatives of (1E)-2-(1-(2-oxo-2H-chromen-3-yl)ethyli-26 dene)hydrazine-1-carbothioamide could be considered promising molecules for potential drug development."

It should be rephrased.

The quality of figures 5 and 6 could be improved.

Wangsa et al. [1] conclude that:

"The specific tyrosinase inhibitor tropolone forms a pre-Michaelis complex with the enzyme. It binds near the binuclear copper site without directly coordinating the copper ions."

Does the same hold for the docked pose of OTR?

"Considering the closeness of the docking score values obtained (43.4042 and 38.2733 for T6 when docked in MTb and MTc sites respectively), it could be assumed that this mixed-type inhibitor might form ternary complexes with the enzyme and the substrate by binding at one of these allosteric sites."

What is the error of the docking scoring function? What is the docking score for MTa?

[1] Crystal Structure of Agaricus bisporus Mushroom Tyrosinase: Identity of the Tetramer Subunits and Interaction with Tropolone

Wangsa T. Ismaya, Henriette J. Rozeboom, Amrah Weijn, Jurriaan J. Mes, Fabrizia Fusetti, Harry J. Wichers, and Bauke W. Dijkstra

Biochemistry 2011 50 (24), 5477-5486

DOI: 10.1021/bi200395t

Reviewer 2 Report

The manuscript by Sebastiano Masuri,et al. entitled “Hydroxylated coumarin-based thiosemicarbazones as dual anti-tyrosinase and antioxidant agents” reports the synthesis of six compounds, five hydroxylated coumarin-based thiosemicarbazones with potent anti-tyrosinase activity and antioxidant properties. The manuscript is well written, but I have a few remarks:

1. Authors had written line 19 - “a panel of new variably hydroxylated derivatives”, line 66 – “new series” or line 538 – “novel compounds with potent anti-tyrosinase activity”. However the T1 (1990), T2 (2022) and T3 (2017) derivatives are known, described in Reaxys database, so only 3 of the compounds are new structures.

2. Authors should include either elemental analysis or a quantitative HPLC/GC trace to support bulk purity. Please note that HRMS is not proof of purity, and NMR silent impurities will not show up in NMR studies.

3. The authors should explain the difference in antioxidant activity between DPPH and ABTS, especially for compounds T1-T3.

3. In the literature, we can find tyrosinase inhibitory activity (IC50) for kojic acid in range 15-20 mM for diphenolase activity and 70-90 mM for monophenolase activity (mushroom tyrosinase), but in presented studies this value is 18 mM. Please explain it.

4. Why the inhibition reaction was monitored following the molecular absorbance at 492 nm, since the formation of the dopachrome product has the absorption maxima at 475 nm?

5. Please explain why compound T1 possess higher inhibitory activity compared to T2 with hydroxyl group.

6. Some misprints need to be corrected:

Line 92 – Reaction instead of reaction (capital letter).

Line 189 – catecholic instead of cathecolic.

Line 638 – MedChemComm instead of Medchemcomm.

The article has been written carefully and the subject is interesting. I really enjoyed reading this article and suggest the acceptance of this article after considering my comments.

Round 2

Reviewer 1 Report

Moderate English changes required

Author Response

Comments and Suggestions for Authors

Moderate English changes required

Answer: we agree.

Correction: The changes to English have been made.